# Rare Event Analysis of Large Language Models

**Jake McAllister Dorman** [1]   **Edward Gillman** [1]   **Dominic C. Rose** [1]   **Jamie F. Mair** [1]   **Juan P. Garrahan** [1]

## Abstract

Being probabilistic models, during inference large language models (LLMs) display *rare events*: behaviour that is far from typical but highly significant. By definition all rare events are hard to see, but the enormous scale of LLM usage means that events completely unobserved during development are likely to become prominent in deployment. Here we present an end-to-end framework for the systematic analysis of rare events in LLMs. We provide a practical implementation spanning theory, efficient generation strategies, probability estimation and error analysis, which we illustrate with concrete examples. We outline extensions and applications to other models and contexts, highlighting the generality of the concepts and techniques presented here.

## 1. Introduction

*Rare events* are occurrences that are both important and atypical in systems whose outcomes are distributed. They are relevant across the sciences, an important example being the processes that mediate physical systems undergoing phase transitions. Three key questions about rare events are the following: (i) how likely are rare events; (ii) what are their properties; and (iii) since they are by definition atypical, what are efficient ways to access and study them? Question (i) concerns the estimation of statistical properties of the rare events, which we thus call *rare event estimation*. Questions (ii) and (iii) concern the study of the structure and properties of the rare events themselves, which we call *rare event exploration*. The answers to these can provide deep physical insights, inform critical safety measures, or guide the design of interventions mitigating negative effects. In traditional statistics, examples include the estimation of rare disease prevalence, the accident rate of commercial aircraft, or the frequency of faults in high volume manufacturing

(McGrath & Burke, 2024). In computational chemistry and statistical physics, powerful techniques for studying rare events have been developed in the context of molecular dynamics. Collectively, we call these questions, studies and mathematical/computational tools *Rare Event Analysis* (REA).

As LLMs become increasingly widespread, the effects of unlikely but important events in the responses they generate become ever more relevant. Yet, rare event analysis for LLMs is still in its infancy (Wu & Hilton, 2025; Jones et al., 2025). Often, in freshly trained LLMs, undesirable outputs occur frequently and are only made improbable through dedicated post-training and guardrails (Mazeika et al., 2024; Lambert et al., 2025). This "alignment" makes the undesirable responses atypical, pushing them toward the tails of the distribution of outputs. However, the scale of current deployment of LLMs is such that events so unlikely as to be almost impossible to observe during model testing will occur with non-negligible frequency when models are released for public use.

**Aims and Objectives:** The primary aim of this work is to provide a *practical* starting point for firmly establishing Rare Event Analysis of LLMs. By practical we mean that the methods and conceptual frameworks presented here can be used broadly by developers, researchers and engineers working on large foundation models, as well as application builders in critical domains, to begin to understand the properties of rare events, and how to control them, in their models and services.

**Framework and outline:** We identify three key stages in the REA of LLMs: (1) **Setup**: defining the rare events to be studied (2) **Estimation**: studying the statistical properties of the rare events, and (3) **Exploration**: studying the structure and properties of the rare events themselves. In what follows, we present a complete end-to-end study covering each stage in turn. A summary of these stages, along with the corresponding sections in which they are covered, is given below:

1. **Setup:** By viewing LLMs as a stochastic process generating outcomes, we can define rare events as atypical values of some "observable" given by a function of the outcomes. We describe this view in Sect. 3.1, and give the specific details of the model and observables used

[1]School of Physics and Astronomy, University of Nottingham, Nottingham, NG7 2RD, UK. Correspondence to: Jake McAllister Dorman <jake.dorman@nottingham.ac.uk>.

*Proceedings of the 43$^{rd}$ International Conference on Machine Learning*, Seoul, South Korea. PMLR 306, 2026. Copyright 2026 by the author(s).

in Sect. 4.

2. **Estimation:** For estimating the probabilities of rare events, a broad literature of methods from statistical physics and computational chemistry exist that can be adapted to LLMs. We present one such approach in Sect. 3, and apply it to our setup in Sect. 5, with some specific experimental details in Sect. 4.

3. **Exploration:** To better predict and control rare model outputs, we need to understand their properties. While this will generally be problem dependent, we demonstrate a generic EDA-like approach in Sect. 6.

To tailor this framework to answer a specific question about a given LLM requires two primary choices. First, a *target observable* i.e. a function of the completion that captures the properties of the model to be studied. Second, a *biasing observable*, i.e. a function of the completion used to influence the sampling distribution in order to highlight rare events. These may naturally be chosen as the same observable, so that the sampling is biased towards rare values of the target observable. However, if the target observable is expensive to calculate, a better biasing observable may instead be a cheaper proxy that is correlated with the target observable. In this work, we will always choose the target and biasing observables to be the same, and use the term observable to refer to both.

For concreteness and practicality, we study the TinyStories model (Eldan & Li, 2023) and ask two primary questions. First, since the model is designed to produce easily readable text by being trained on children's stories, we ask: how often does this model produce completions that are very difficult to read? To answer this, we take the automated readability index (ARI) (Kincaid et al., 1975), a well-known linguistic measure of text complexity, as our observable. Second, we ask a more generic question: how often does the model produce completions with very low or very high probabilities? For this, we use the natural observable of the log-probability of a completion. These choices allow for a clear illustration of how REA can be applied to LLMs, without requiring the specialist domain knowledge required for other problems e.g. in AI safety evaluation. Throughout, we provide practical details on theory, on sampling (i.e., generation) strategies, probability estimation and error analysis.

In general, REA presents significant challenges. A systematic exploration of atypical behaviour in a stochastic system is computationally demanding in the best of cases, and can become prohibitive for large or slow systems. This is certainly the case when performing REA in LLMs. It is therefore important when describing an end-to-end analysis such as the one presented here to make clear the technical limitations at every stage. This also demands a balance between

theoretical rigour and practical implementation details, and acknowledging the limitations of accessible computational resources. One approach would be to only consider a fully solvable toy problem which might allow an exhaustive theoretical analysis, at the expense of oversimplification and lack of practical relevance. At the other end, one could consider only a large scale LLM which might correspond to a realistic scenario but which, due to its scale, would prevent us from demonstrating the REA techniques. By utilizing TinyStories here, we aim for a mid-point between these two extremes that demonstrates the theory and its application in a scenario representative of reality. In our view this is the best way to achieve our primary aim of providing a practical starting point for the field by demonstrating an end-to-end implementation of REA. As will be evident below, the ultimate implementation for industry standard LLMs will require the widespread collaboration of experts and deep technical contributions across fields.

**Summary of Contributions:** (1) We present for the first time a complete end-to-end application of rare event analysis to LLMs. (2) We provide a practical guide to implementation of these methods including theory, generation strategies, probability estimation and error analysis. (3) We analyse the probabilities of rare completions in TinyStories models for two observables of practical interest: ARI and Log-Probability. (4) We provide an example of exploratory data analysis (EDA) of the realisations of these rare completions. (5) We outline extensive directions for improvements to the algorithms, and applications to other models and contexts.

For reproducibility, we provide a minimal code implementation of our framework here (McAllister Dorman et al., 2026).

## 2. Relation to Other Work

In the context of physical sciences, a number of techniques have been developed to study rare events, particularly for estimating the free energy in molecular dynamics simulations (Frenkel & Smit, 2023). This includes methods such as umbrella sampling (Vardi, 1985; Torrie & Valleau, 1977; Thiede et al., 2016), thermodynamic integration (Kirkwood, 1935; Gelman & Meng, 1998), and the Multistate Bennett Acceptance Ratio (MBAR) estimator (Shirts & Chodera, 2008; Shirts, 2017) which we use in this work. Such methods rely on combining samples from multiple biased simulations to improve estimates. The samples themselves are typically generated using Markov Chain Monte Carlo (MCMC) methods (Andrieu et al., 2003; Brooks et al., 2011) such as Metropolis-Hastings (MH) (Metropolis et al., 1953; Hastings, 1970), often applying techniques such as simulated annealing (Bertsimas & Tsitsiklis, 1993), parallel tempering or replica exchange (Earl & Deem, 2005) to improve exploration of the state space. The version of MH we consider

is Transition Path Sampling (TPS) (Bolhuis et al., 2002; Hedges et al., 2009), originally designed for sampling trajectories of dynamical systems. While MCMC is standard, there are a number of important alternative sampling methods such as Sequential Monte Carlo (SMC) (Doucet et al., 2001; Cérou et al., 2012) and particle filtering or cloning (Giardina et al., 2011).

Error analysis for rare event probabilities is an active area of research. This includes work on standard sampling problems (Dembinski & Schmelling, 2022; McGrath & Burke, 2024), MCMC methods (Flegal & Jones, 2010; Rosenthal, 2018; Jiang et al., 2022), and the application of MBAR to correlated data (Dinner et al., 2020; Li et al., 2023; Ding, 2024). Here, we use bootstrap methods to construct confidence intervals (CIs) for MBAR estimates (Davison & Hinkley, 1997) and Wilson intervals for direct sampling (Wilson, 1927).

Regarding rare event probability estimation for LLMs, the most closely related works are those of (Wu & Hilton, 2025) and (Jones et al., 2025). The former focuses on the limited setting of estimating the probability of generating a single rare token by varying the prompt, while here we consider full completions and more general observables. The latter work shows how for a production LLM (Claude 3/3.5) rare event probability estimates on a small set of test prompts can be extrapolated to a much larger set of deployment prompts. The results of (Jones et al., 2025) give an important practical justification for the work we present here: we focus on accurate rare event probability estimation and error quantification for single/few prompt settings. We show how, even with strong resource limitations, detailed estimates of rare event probabilities can be obtained by applying rare event sampling methods. The results of (Jones et al., 2025) then suggest that such detailed few-prompt estimates can be effectively combined with extrapolation methods to provide accurate rare event probabilities in practical deployment settings. While both these works focus on the general problem of rare events in LLMs, neither provide methods that are directly comparable to the questions we tackle here. Indeed, the only directly comparable method for our setup is direct sampling of the LLM, which is thus the current "state-of-the-art" for rare completion probability estimation.

In addition to the work on rare event estimation for LLMs, a number of works in related fields exist. Both SMC (Lew et al., 2023; Loula et al., 2025) and TPS (Faria et al., 2024; Faria & Smith, 2025) have been applied previously for constrained/biased generation, while in the field of extreme value theory, initial connections have been made to response lengths in LLMs (Jiao et al., 2025). More broadly, the Reinforcement Learning from Human Feedback (RLHF) (Bai et al., 2022) and Direct Preference Optimization (DPO) (Rafailov et al., 2023) methods for alignment share objec-

tive functions to those used in rare event sampling, namely that of a time-integrated "reward" function with a Kullback-Leibler (KL) divergence regularisation. This provides a further link to a variational perspective (Chetrite & Touchette, 2015; Jack & Sollich, 2015) for REA that has been approached with reinforcement learning (RL) (Rose et al., 2021; Das et al., 2021; Gillman et al., 2024; Pamulaparthy & Harris, 2025).

Finally, we note that recent work has also focused on "distribution sharpening" as a means to increase the likelihood of sampling high probability outputs from LLMs. The approach used in (Karan & Du, 2025) is directly analogous to TPS, and specifically the "shooting method" (see below), while the approach in (Ji et al., 2026) estimates and samples from a modified distribution known as the "Doob transform" (Chetrite & Touchette, 2015; Jack & Sollich, 2015; Garrahan, 2016). This underlines the usefulness of applying well known rare event techniques from statistical mechanics to LLMs.

## 3. Background on Rare Event Methods

### 3.1. Language Models as Stochastic Processes

Suppose we have a prompt $\mathbf{x}_{1:t} = (x_1, \ldots, x_t)$ consisting of $t$ tokens $x_i$. An LLM produces a distribution over all tokens in the vocabulary of the model, equal to the probability that the token follows the given prompt. We denote the corresponding probability mass function (PMF) as $p_{\mathcal{M}}(x_{t+1}|\mathbf{x}_{1:t})$. A *completion* to this prompt, $\mathbf{x}_{t+1:T} = (x_{t+1}, \ldots, x_T)$, is then generated by sequentially sampling from this distribution. The probability of this completion is then given by,

$$p_{\mathcal{M}}(\mathbf{x}_{t+1:T}|\mathbf{x}_{1:t}) = \prod_{\tau=t}^{T-1} p_{\mathcal{M}}(x_{\tau+1}|\mathbf{x}_{1:\tau}). \qquad (1)$$

The form (1) is sometimes called *autoregressive* and we refer to the sequence $\mathbf{x}_{1:T}$ as a trajectory.

Since it is always fixed during generation, for brevity we will suppress explicit conditioning on the prompt from now on, except when useful for clarity.

### 3.2. Importance Sampling

Consider an expectation under the distribution of the model,

$$\bar{f} := \mathbb{E}_{p_{\mathcal{M}}}[f(\mathbf{x}_{1:T})] = \sum_{\mathbf{x}_{t+1:T}} f(\mathbf{x}_{1:T}) p_{\mathcal{M}}(\mathbf{x}_{t+1:T}),$$

$$= \sum_{x_T} \cdots \sum_{x_{t+1}} f(\mathbf{x}_{1:T}) p_{\mathcal{M}}(\mathbf{x}_{t+1:T}|\mathbf{x}_{1:t}). \qquad (2)$$

If $f$ is a binary function such as,

$$f(\mathbf{x}_{1:T}) = \mathbb{I}[\phi(\mathbf{x}_{1:T}) \in \mathcal{D}], \qquad (3)$$

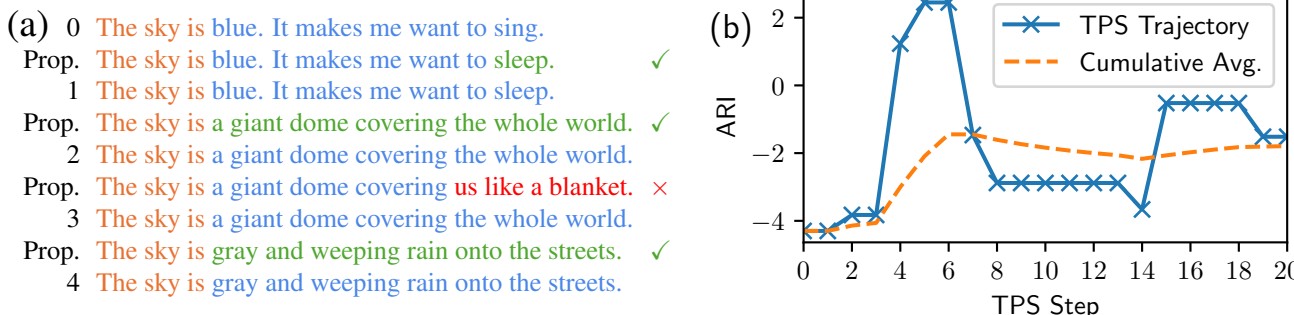

*Figure 1.* **(a) Text generation**: Shown is a single "trace" of the text produced by the TPS text generation process. The prompt (orange) remains fixed throughout, while the completion (blue) varies. At each step an edit to the completion is proposed that is either accepted (green), or rejected (red), leading to no change. **(b) Evolution of the observable in a TPS trajectory**: Automated readability index (blue, see Sec. 4) and its cumulative average (orange, dashed) along the TPS trajectory shown in (a).

for some observable $\phi$, set $\mathcal{D}$, and where $\mathbb{I}(\cdot)$ is the indicator function taking value 1 if the condition is true and 0 otherwise, then $\bar{f}$ is the probability that the observable takes values in the set $\mathcal{D}$. If $p_{\mathcal{M}}$ assigns sufficiently small probabilities to values $\mathbf{x}_{1:T}$ where $f(\mathbf{x}_{1:T}) = 1$, then an empirical estimate of Eq. (2) will almost certainly give zero.

To remedy this, suppose instead that we have a distribution with PMF $p^*$ that gives much greater probability mass to regions where $f = 1$. We can then instead evaluate (2) as,

$$\bar{f} = \mathbb{E}_{p^*}\left[w(\mathbf{x}_{1:T})f(\mathbf{x}_{1:T})\right], \qquad (4)$$

where now samples are taken from $p^*$ and weighted by

$$w(\mathbf{x}_{1:T}) := \frac{p_{\mathcal{M}}(\mathbf{x}_{t+1:T}|\mathbf{x}_{1:t})}{p^*(\mathbf{x}_{t+1:T}|\mathbf{x}_{1:t})}. \qquad (5)$$

This is known as *importance sampling*.

Importance sampling can also be applied when $p^*$ is a mixture distribution,

$$p^*_{\text{Mix}}(\mathbf{x}_{t+1:T}) := \sum_{k=1}^{K} \alpha_k p^*_k(\mathbf{x}_{t+1:T}), \qquad (6)$$

where the components, $\{p^*_k\}_{k=1}^K$, are weighted by $\{\alpha_k\}_{k=1}^K$ with $\sum_k \alpha_k = 1$ and $\alpha_k \geq 0 \, \forall \, k$. Setting $p^* = p^*_{\text{Mix}}$ in Eqs. (4) and (5) and denoting the corresponding weight as $w_{\text{Mix}}$ gives,

$$\begin{aligned}
\bar{f} &= \mathbb{E}_{p^*_{\text{Mix}}}\left[w_{\text{Mix}}(\mathbf{x}_{1:T})f(\mathbf{x}_{1:T})\right], \\
&= \sum_{k=1}^{K} \alpha_k \mathbb{E}_{p^*_k}\left[w_{\text{Mix}}(\mathbf{x}_{1:T})f(\mathbf{x}_{1:T})\right], \qquad (7) \\
&= \sum_{k=1}^{K} \alpha_k \mathbb{E}_{p^*_k}\left[\frac{p_{\mathcal{M}}(\mathbf{x}_{t+1:T}|\mathbf{x}_{1:t})}{\sum_{j=1}^{K} \alpha_j p^*_j(\mathbf{x}_{t+1:T}|\mathbf{x}_{1:t})}f(\mathbf{x}_{1:T})\right]. \\
& \qquad\qquad\qquad\qquad\qquad\qquad\qquad\qquad (8)
\end{aligned}$$

This is often referred to as umbrella sampling (Torrie & Valleau, 1977; Dinner et al., 2020; Li et al., 2023).

### 3.3. Exponentially Reweighted Distributions

Whilst any distribution with a greater or equal support to that of $p_{\mathcal{M}}$ can be used for importance sampling, we will consider the *biased* or *tilted distribution* defined by exponential reweighting according to the trajectory observable $\phi$. This has the PMF,

$$p_\lambda(\mathbf{x}_{t+1:T}) := \frac{1}{Z(\lambda)} e^{-\lambda\phi(\mathbf{x}_{1:T})} p_{\mathcal{M}}(\mathbf{x}_{t+1:T}), \qquad (9)$$

where $\lambda$ is the *bias* (or *tilting* parameter), and $Z(\lambda)$ is a normalisation constant commonly referred to as the *partition function*. By changing $\lambda$ this choice of distribution encourages sampling of rare events while keeping the samples generated representative of the original model. In statistics, this is known as the exponential family and has a number of important properties (Murphy, 2022). Normalisation of $p_\lambda$, i.e. requiring $\mathbb{E}_{p_\lambda}[1] = 1$, gives,

$$Z(\lambda) = \sum_{x_T}\cdots\sum_{x_{t+1}} \exp[-\lambda\phi(\mathbf{x}_{1:T})]p_{\mathcal{M}}(\mathbf{x}_{t+1:T}), \qquad (10)$$

$$= \mathbb{E}_{p_{\mathcal{M}}}\left[e^{-\lambda\phi(\mathbf{x}_{1:T})}\right]. \qquad (11)$$

For large $T$ or high dimensional data, exact calculation of $Z(\lambda)$ is often impossible. $Z(\lambda)$ must then be treated as unknown and estimated.

To use these distributions with umbrella sampling, we define a mixture with components setting $p^*_k = p_{\lambda_k}$ for $K$ values of the bias, $\{\lambda_k\}_{k=1}^K$. This gives a distribution where $p_{\mathcal{M}}$ factorises,

$$p^*_{\text{Mix}}(\mathbf{x}_{t+1:T}) = p_{\mathcal{M}}(\mathbf{x}_{t+1:T}) \sum_{k=1}^{K} \alpha_k \frac{e^{-\lambda_k\phi(\mathbf{x}_{1:T})}}{Z(\lambda_k)}. \qquad (12)$$

This then gives a form for $w_{\text{Mix}}$ in Eq. (8) where the factor

of $p_{\mathcal{M}}$ in the numerator cancels,

$$w_{\text{Mix}}(\mathbf{x}_{1:T}) = \frac{1}{\sum_{j=1}^{K} \alpha_j Z^{-1}(\lambda_j) e^{-\lambda_j \phi(\mathbf{x}_{1:T})}} \ . \qquad (13)$$

This depends on the $K$ unknown partition functions $Z(\lambda_j)$, which must therefore be estimated in order to evaluate the expectation (8). Starting from the definition of $Z(\lambda_j)$ as an expectation in (11), we then apply the umbrella sampling formula (8) setting $f(\mathbf{x}_{1:T}) \to \exp\left(-\lambda_j \phi(\mathbf{x}_{1:T})\right)$ to give $K$ simultaneous equations for the partition functions,

$$Z(\lambda_j) = \sum_{k=1}^{K} \alpha_k \mathbb{E}_{p_{\lambda_k}} \left[ w_{\text{Mix}}(\mathbf{x}_{1:T}) e^{-\lambda_j \phi(\mathbf{x}_{1:T})} \right] \ . \qquad (14)$$

Since the right-hand side of (14) depends on all $\{Z(\lambda_j)\}_{j=1}^{K}$ through $w_{\text{Mix}}$, these equations must be solved self-consistently along with (8) to obtain estimates of the partition functions and the desired expectation.

In practice, one applies MCMC to obtain samples from the distributions in these equations, and the resulting estimates are consistent estimates for any choice of $\alpha_k$ (Dinner et al., 2020). Choosing $\alpha_k = N_k^{-1}$, where $N_k$ is the number of samples obtained from $p_{\lambda_k}$, is optimal when the samples are uncorrelated (Shirts & Chodera, 2008). The resulting method is known as MBAR.

### 3.4. Monte Carlo for Sequences and Transition Path Sampling

Metropolis-Hastings provides a general method to sample from a target distribution, $p^*$, by constructing a Markov chain (MC) whose stationary distribution is $p^*$. This is achieved by sequentially proposing new states through a *proposal distribution function*, $p_{\text{Prop}}(\mathbf{x}_{\text{new}}|\mathbf{x}_{\text{old}})$, and accepting these proposals with a probability determined by an *acceptance function*, $A(\mathbf{x}_{\text{new}}, \mathbf{x}_{\text{old}})$. If the proposal is accepted, the state is set to the proposal and is otherwise unchanged. A single iteration of proposal and acceptance/rejection gives one "step" of the MC. The MH approach determines a form for the acceptance function such that the resulting MC dynamics obeys *detailed balance* with respect to $p^*$, meaning it is guaranteed to sample from $p^*$ when run for a sufficient number of steps. The average of samples from this MC then gives an almost-sure and asymptotically normal estimator for $\bar{f} = \mathbb{E}_{p^*}[f(\mathbf{x})]$ (Brooks et al., 2011).

When the objects of interest are ordered sequences, as is the case for LLMs, a convenient variant of MH is TPS (Bolhuis et al., 2002). In particular, suppose we are on step $i$, with current state, $\mathbf{x}_{1:T}^{(i)} = (x_1, \ldots, x_T)$. TPS generates a proposal by first truncating this trajectory at some index $\tau \in [1, T)$ and then autoregressively sampling from some distribution to generate a new trajectory, $\tilde{\mathbf{x}}_{1:\tilde{T}} = (x_1, \ldots, x_{\tau-1}, \tilde{x}_\tau, \ldots, \tilde{x}_{\tilde{T}})$, see Appendix B for

details. This process is shown in the context of LLMs in Fig. 1.

## 4. Experimental Setup

As a concrete study of rare event analysis for LLMs, we consider the TinyStories-8M model. By training on LLM-generated children's stories with limited vocabulary but sophisticated concepts, the TinyStories models are capable of producing human-like text with only a few million parameters (Eldan & Li, 2023). This makes them ideal for initial developments of REA with sampling-based methods and, even in a resource-limited setting, we are able to generate millions of samples. When using direct sampling, such quantities are sufficient for statistical analysis of distributions close to typicality. However, to explore the tails of the distribution and thus rare events, specialised sampling methods are required. To give an idea of the computational cost required, the histograms we report in Sect. 5 require of the order of $10^9$ tokens to be generated. For comparison, for its flagship LLM Gemini, Google reports processing of the order of $10^{15}$ tokens per month (Hassabis, 2025) so that equivalent histograms for Gemini would require about one second of their total worldwide processing.

We consider completions from the TinyStories-8M model generated using direct / ancestral sampling, that is "temperature 1.0" decoding (Shi et al., 2024). All completions consist of 100 tokens generated from the fixed prompt: "*Once upon a time, in a big forest, there lived a rhinoc*", consisting of the first 16 tokens of an entry in the validation split of the TinyStories dataset. Comparative results for a range of prompts are provided in Appendix D. Rare events are then quantified in terms of the extreme values of two observables of interest computed over the prompt-completion pair.

The first observable is the logarithm of the joint probability (Log-Prob) for the completion given the prompt. This provides an intrinsic measure of how likely a given completion is for a pre-trained model under a "temperature 1" decoding strategy. It thus yields important insights into the model itself. Extreme values of the Log-Prob correspond to completions that are either very likely/unlikely, with the former being a common focus for the development of decoding algorithms (Shi et al., 2024).

The second observable is the automated readability index (ARI) (Kincaid et al., 1975). This is a standard linguistic metric of readability defined as

$$\text{ARI}(x) = 4.71 \cdot \frac{c(x)}{w(x)} + 0.5 \cdot \frac{w(x)}{s(x)} - 21.43, \qquad (15)$$

where $x$ denotes the input text, and $c(x), w(x)$ and $s(x)$ are the number of characters, words and sentences in $x$, respectively. The ARI quantifies the expected reading age

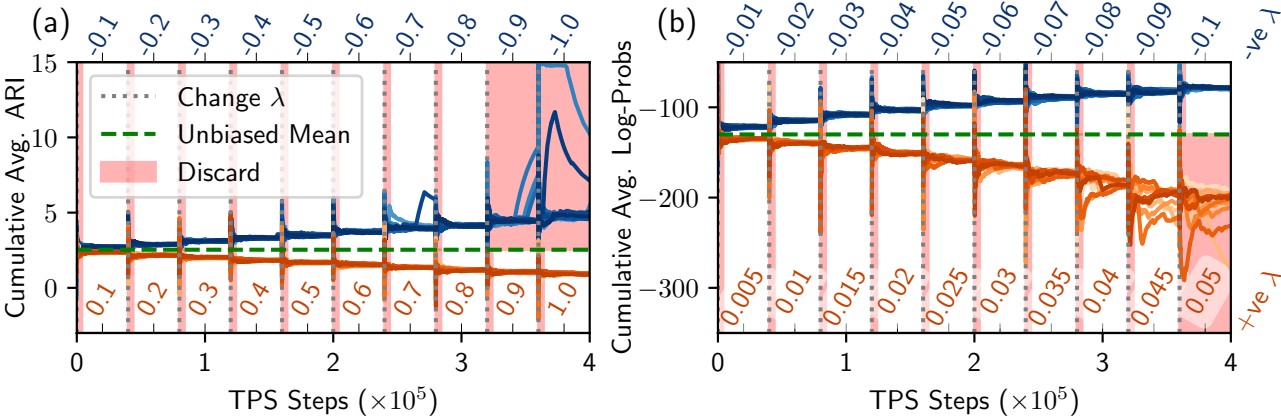

*Figure 2.* **Observables in annealing TPS trajectories**. (a) Cumulative average of the ARI along TPS trajectories for the TinyStories LLM. We show both positive (orange) and negative (blue) biases, generated as described in Sect. 4. The cummulative average resets when the bias changes. The annealing schedule consists of 10 values of the bias increasing in magnitude, with each bias being run for $4 \times 10^4$ TPS steps, totaling $4 \times 10^5$ samples per bias. The first 10% of each chain is discarded as "burn-in", and we discard all samples from any bias with a GR statistic of greater than 1.1 to avoid non-convergence (thus discarded samples are shown by the shaded red regions). The mean value of the ARI for the unbiased "temperature 1" distribution is shown by the green dashed line. On average 50 tokens are generated per step of TPS, resulting in approximately $4 \times 10^8$ tokens generated. (b) Same for the Log-Probs observable.

required to understand a text: an ARI of 1 corresponds with ages five to six, while an ARI of 14 corresponds with a university age student. Related measures are used as part of regulatory guidance / requirements for communications in some countries, c.f. Chapter 8 of Ref. (FCA, 2022), making extreme readability scores of practical interest. Given that the TinyStories models are trained to produce text aimed at children, completions with high ARI scores should be both unlikely and can be considered as an example of "unwanted" behavior for this model, i.e. a failure in its "alignment". We compute the ARI over the entire prompt-completion pair, with a cap at 15. This is linguistically motivated, since in common use ARI is taken as an integer between 1 and 14, rounded down. We also found that this cap improved the performance of the MCMC by mitigating the impact of very rare completions with both very high ARI values and high Log-Prob values that lead to very low acceptance rates.

We compare completions generated through two methods: direct sampling and TPS. Using direct sampling, the current "state-of-the-art" for rare completion probability estimation, we generate 4.2 million completions, corresponding to approximately 420 million tokens generated from the model per observable. These are i.i.d. samples of the distribution of interest.

Direct sampling is inefficient, as it generates many more samples than needed for typical values, and too few (or no) samples for atypical values, preventing the exploration of the tails of the distribution. More efficient is to use TPS by targeting the exponentially tilted distribution (9) using the observable of interest for $\phi$ for different values of $\lambda$, and

then reconstructing the original distribution.

We do this by considering both positive and negative biases $\lambda$ to target both tails of the distribution. Each of these are connected in two separate chains under TPS dynamics via an "annealing" protocol, which gradually increases the magnitude of the bias after a fixed number of steps, see e.g. (Mair et al., 2023). This aims to improve convergence by gradually changing the bias so that each fixed bias TPS segment starts closer to its target distribution. Fig. 2 displays the corresponding running averages of the chains, and illustrates the schedule used.

To construct the estimates for the rare event probabilities in Sect. 5 below we use MBAR via the *pymbar* package (Shirts & Chodera, 2008; Beauchamp et al.) to combine the TPS samples with an additional $2 \times 10^5$ directly sampled completions. This corresponds to a total exceeding $4 \times 10^8$ tokens generated for each observable, matching the number of tokens generated through direct sampling and allowing for a fair comparison of the two methods. We take steps that are routine in more standard MC simulations, such as preprocessing the TPS samples by discarding the first 10% of each TPS trajectory as "burn-in" (to ensure the remaining TPS samples are representative of the target distribution), and use the Gelman-Rubin (GR) statistic as a convergence diagnostic, removing all samples where GR $\geq$ 1.1 (Roy, 2020). See Appendix C.3 for details.

For the rare completion exploratory data analysis presented in Sect. 6 below, we generate an additional $2.1 \times 10^6$ completions using TPS with the same annealing schedule for the bias $\lambda$, but with a reduced $2 \times 10^4$ steps per bias.

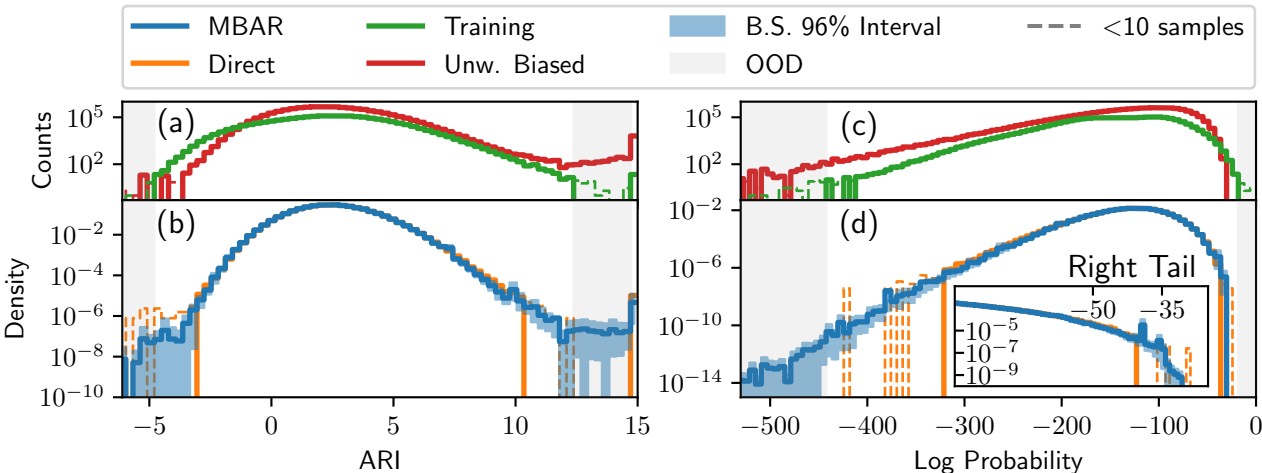

*Figure 3.* **Distributions of observables for the TinyStories-8M model**. (a) Number of counts (red) for values of the ARI from all the biased simulations. This is the raw input to MBAR for reconstructing the true distribution. For comparison, we show the distribution of the ARI in the training data (green). The shaded areas indicate values of ARI with fewer than 10 samples in the training data. (b) Inferred normalised density (blue) of the ARI, using MBAR with the accepted samples generated through TPS in Figure 2, plus an additional $2 \times 10^5$ samples from direct sampling (totalling about $7 \times 10^6$ out of about $8 \times 10^6$ generated completions, on average 50 tokens generated per completion). The shaded areas demarcate the 96% CI. For comparison we also show the corresponding distribution of ARI (orange) from direct sampling ($4.1 \times 10^6$ completions, 100 tokens per completion). (c,d) Same for Log-Probs.

## 5. Rare Completion Probability Estimation

Denoting the marginal probabilities for $\phi(\mathbf{x}_{1:T})$ to reside in the interval $s_l = [a_l, a_{l+1}]$ as $\mathbb{P}(\phi(\mathbf{X}_{1:T}) \in s_l)$, this can be estimated via the expectations,

$$\mathbb{P}(\phi(\mathbf{X}_{1:T}) \in s_l) = \mathbb{E}_{p_\mathcal{M}} \left[ \mathbb{I}[\phi(\mathbf{x}_{1:T}) \in s_l] \right] . \qquad (16)$$

Dividing by the interval widths gives a histogram representation of the marginal probability density, allowing for visualisation and analysis of the tails of the distribution. In the limit of infinitely many, infinitesimally small intervals, this recovers the full marginal density (Dinner et al., 2020).

The resulting histograms are displayed in Fig. 3. While the direct histogram contains only a few samples in the tails, with many bins containing no samples at all, the MBAR histogram contains counts across the range, and allows for probing densities orders of magnitude smaller than accessible via direct sampling. For the ARI, while the direct histogram is limited to the range of the training data, the MBAR histogram extends well beyond this, allowing for insights into how the model generalises to these regions. We note that these distributions are strongly non-Gaussian, a common trait when correlations are present, as here due to sequential token generation, leading to increased likelihood of rare outcomes.

Interval estimates for each expectation are constructed as 96% confidence intervals (CIs) using percentile bootstrap on the independent simulation chains (Davison & Hinkley, 1997). Given $M$ TPS trajectories, we sample with replace-

ment $M$ times to construct a single "bootstrap replica". For each replica we recompute the density estimates (including preprocessing with burn-in and GR). Taking 100 replicas total, the smallest/largest third percentile of the estimates give the lower/upper bound of the CI shown as shaded areas in Fig. 3.

The very low values of probability estimates for rare events mean that a key evaluation of estimate quality is the widths of CIs relative to the estimated value (McGrath & Burke, 2024). These are displayed for the two histograms in Fig. 4(a), with the MBAR histogram CIs as described above and a 96% CI based on the Wilson interval (Wilson, 1927) used for the direct histogram. As can be seen, in the tails the MBAR histogram displays significantly smaller relative CI widths than the direct histogram, although rigorous interpretation in terms of coverage probabilities requires care (McGrath & Burke, 2024).

Fig. 4(b) displays analysis of the change in statistical bias. The top row compares the final estimates with the change in bin height estimates when doubling the number of MCMC steps, to better sample the target distributions and reduce bias. While the ratio is generally small, some are close to 1 in the tails. This indicates that while the results in the tails may change substantially with more steps, they are likely of the right order of magnitude.

To improve estimates, we may either increase the steps per MCMC chain or run more parallel chains. While the former reduces both bias and variance, the later only reduces

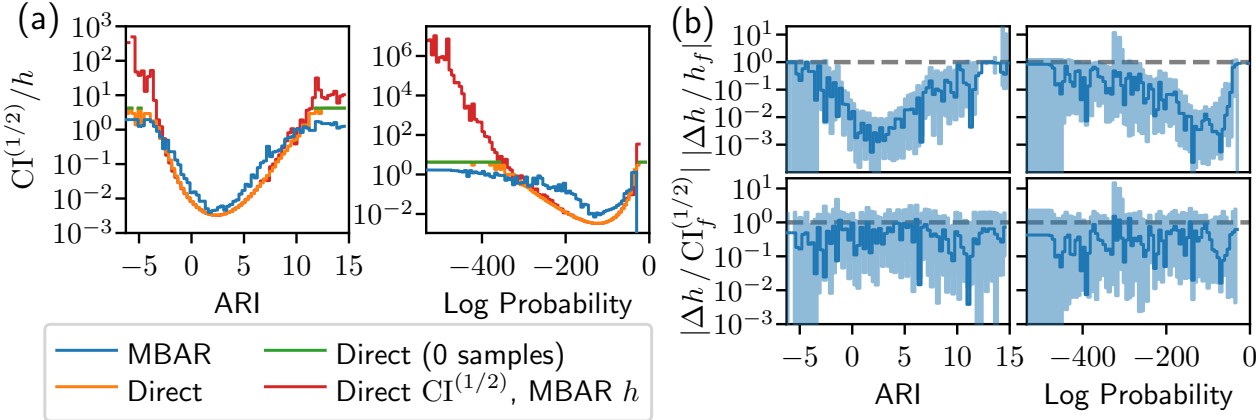

Figure 4. **Error analysis**. (a) Relative half width CI CI$^{(1/2)}$ for the MBAR (blue) and direct (orange) estimates, for ARI (left panel) and Log-Probs (right panel). To allow for comparison, two estimates for the heights of bins with no counts from direct sampling are used: half the height of the smallest non-zero bin from the direct sampling (green), and the heights from the MBAR estimate (red). The latter represents our best guess for the true bin heights in the tails, and the relative error shows the dramatic improvement provided by the MBAR estimate. (b) Difference between the bin heights, $\Delta h$, using the full TPS trajectories, $h_f$, and the height using only the first half (per bias in the annealing schedule) of the TPS trajectories, $h_h$, after burn-in. The change is normalised in two ways: by the histogram heights computed using the full dataset (top), and by the confidence interval half-width CI$^{(1/2)}$ (bottom).

variance. To determine which to increase, we compare the change in bias to the CI half-widths in the bottom row of Fig. 4(b), finding the change varies from comparable to slightly smaller than the CI half-widths. This suggests that here focusing on increasing steps within each MCMC chain could lead to a larger reduction in statistical error, as both statistical bias and variance contribute to the overall error, rather than increasing the number of independent chains.

## 6. Rare Completion Exploration

While the methods for rare event probability estimation in Sect. 5 are broadly applicable across domains, rare event exploration, i.e. the analysis of concrete realisations of rare events, is highly domain dependent. For example, in AI safety studies, one might generate dangerous completions to aid the development of specialised guardrails and filters. To present a useful analysis here applicable across domains, we focus on EDA techniques applied to rare completions. Such techniques provide an important starting point for any exploratory analysis of rare completions, upon which domain-specific investigations can build.

As an example application, it may be desirable to filter out rare completions with extreme values of the observable of interest, e.g. to prevent unsafe responses. However, such filtering may be difficult if the observable requires a complete generation to be evaluated, or is expensive to compute. One solution is to identify cheaper-to-compute proxies for the observable, which can be monitored at runtime to enable early, cheap filtering.

Here, we demonstrate how EDA techniques can be used to identify such proxies, using high ARI completions generated by the TinyStories-8M model as a case study. Since the ARI requires complete, multi-sentence strings to be accurately computed, finding useful proxies is of practical interest for filtering out high ARI completions.

The proxy we will consider here is the number of consecutive token repeats, $\text{Repeats}(\mathbf{x}_{1:T}) = \sum_{t=1}^{T-1} \mathbb{I}[x_{t+1} = x_t]$. This choice is motivated first by our observation of highly repetitive text in extreme ARI completions (see below), second by the fact that it is of interest more generally e.g. for decoding strategies (Holtzman et al., 2020) and third because it is implemented efficiently in many LLM libraries, making it a practical candidate for a proxy.

Taking samples corresponding to a large bias towards high ARI, Fig. 5 compares the resulting ARI scores against log probabilities and consecutive token repeats. While the bulk of the distribution shows a weak negative correlation between ARI and Log-Probability, as expected from the training data, there is a clear excursion of high ARI completions, well beyond the regime of the training data but nonetheless having high log probabilities. These completions thus represent a form of extrapolation, and their characterisation can provide insights into the model's learnt rules. For instance, as can be seen from the colour map and inset, these completions are often highly repetitive, with one completion reading: "*eros. He had a friend named Trurururu. Trururururururu was very slow, so the other rhinrururururururururururu complained to Trururururururu...* ['*ru*' continues 50 more times]". Despite the fact that this text is out of dis-

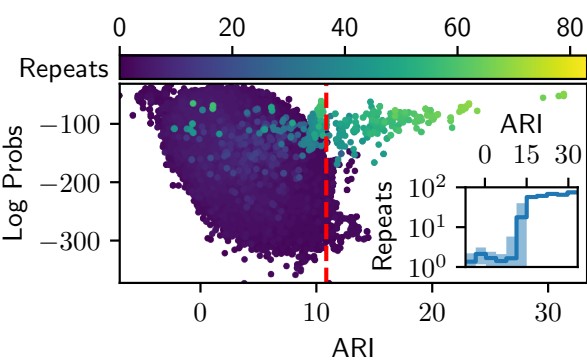

*Figure 5.* **ARI versus Log-Probs**. The ARI and Log-Probs of samples generated by biasing towards ARI, with the colour representing the number of consecutive token repeats. The 0.9999 quartile for the ARI in the training data is shown by the red, dashed line. Inset: Mean (line) and standard deviation (shaded) of the repeat counts per bin, where bins are determined by ARI scores.

tribution with respect to the training data and would never be produced by a human, it is still assigned a high probability by the model. This suggests that, when extrapolating outside the regime of the training data, the model falls back on basic patterns that favour repetition to yield acceptable likelihoods.

## 7. Conclusion and Outlook

We have presented a complete framework to analyse rare events in LLMs. Using this framework we have demonstrated how to estimate the probability of completions that are orders of magnitude less likely than those detectable by direct sampling, the current "state-of-the-art" for this problem. We have shown how to achieve this using efficient sampling techniques that overcome the prohibitive computational cost of direct sampling. We have also shown that by accessing rare completions systematically one can probe model behavior in out-of-distribution regimes.

While we have focused on a class of small LLMs with open weights, this is not a limitation. Firstly, the theory and mathematics behind the framework and methods presented apply regardless of model size. Secondly, our results show that the computational costs (in token-budgets) of the analysis presented can already be extended to larger LLMs at a cost that is modest compared to production usage and training budgets. Therefore, especially with algorithmic improvements (see below) the methods here could be directly employed for critical applications such as safety analysis for large production models by developers.

There are many potential algorithmic improvements possible, e.g. by adapting methods from statistical mechanics. These include: adaptive run-times based on convergence diagnostics (Roy, 2020), parallel tempering (replica exchange)

(Earl & Deem, 2005), and optimisation of biases/samples/M-BAR weights (Klimovich et al., 2015; Li et al., 2023). More sophisticated MCMC proposal distributions such as infilling (i.e. stochastic bridges) (Bavarian et al., 2022; Mair et al., 2023) and fine-tuned models (Rose et al., 2021) would improve performance when considering longer generations. Additionally, many improvements to statistical error estimation are possible with sophisticated bootstrap methods (Davison & Hinkley, 1997), Bayesian approaches (Ding, 2024), and asymptotic analysis (Li et al., 2023).

Regarding model weights, we further emphasise that the approach presented here needs only completions for the model being studied. This allows for the application of such methods without the need for developers of foundation models to provide proprietary model weights and, correspondingly, for researchers to perform rare event analysis accessing models only via standard APIs. If a more complex choice is made for the MCMC proposals, or alternative distributions are sampled from for use in importance sampling, then the model's token log probabilities for these completions will also be required.

Looking ahead to REA for production LLMs, it will be important to consider the variability of user prompts. Promisingly, recent work has shown that rare event properties in a deployment setting with many diverse prompts can be extrapolated from only a few test prompts (Jones et al., 2025), and such extrapolation could be combined with the methods presented here. Additionally, fine-tuning proposal models for a wide range of prompts would substantially reduce costs relative to tackling each prompt independently. Beyond probability estimation, rare event methods can be used to discover prompts associated with rare outcomes (Wu & Hilton, 2025). This has applications to automated red teaming, or for finding prompts that give high-quality completions. Rare event methods can also probe out-of-distribution behaviour to analyse how LLMs extrapolate beyond training data, potentially revealing underlying rules governing generation or highlighting failure modes (Yang et al., 2024; Lu et al., 2025). Finally, we note that in many practical applications, the choice and construction of biasing functions will be a major consideration. This is familiar from the use of reward models in alignment, and will be particularly important for REA when the properties of interest are sparsely non-zero. In those cases, effective REA requires constructing smoothly varying proxies that correlate well with the property of interest, for example via domain-expert insights or reward models.

## Acknowledgements

We are grateful to Jack Paine and Chaitanya Manem for assistance with infrastructure and research developments in the early exploratory stages of this project. We acknowledge

support by His Majesty's Government. We acknowledge partial support from EPSRC Grant No. EP/V031201/1 (J.P.G.). We are grateful for access to the University of Nottingham's Ada HPC service.

## Impact Statement

This paper presents work whose goal is to advance the field of machine learning. There are many potential societal consequences of our work, none of which we feel must be specifically highlighted here.

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

## A. List of Acronyms

- ARI = automated readability index.

- CI = confidence interval.

- DPO = direct preference optimization.

- EDA = exploratory data analysis.

- GR = Gelman-Rubin.

- KL = Kullback-Leibler.

- LLM = large language model.

- Log-Prob = logarithm of the joint probability of a completion.

- MBAR = multistate Bennett acceptance ratio.

- MC = Monte Carlo.

- MCMC = Markov chain Monte Carlo.

- MH = Metropolis-Hastings.

- PMF = probability mass function.

- REA = rare event analysis.

- RLHF = reinforcement learning from human feedback.

- SMC = sequential Monte Carlo.

- TPS = transition path sampling.

## B. Algorithms for Markov Chain Monte Carlo and Transition Path Sampling

### B.1. Preliminaries: Markov Chains

Consider a discrete-time, time-homogeneous Markov chain $\boldsymbol{X} = \{X(t) : t = 0, 1, 2, \ldots\}$ with a fixed, finite state space $\mathcal{X}$ and transition matrix,

$$[\mathbf{P}]_{ij} = \mathbb{P}(X(t+1) = x_i | X(t) = x_j) = p(x_i|x_j)\,, \tag{17}$$

where $x_i, x_j \in \mathcal{X}$ are states in the state space and $p(x_i|x_j)$ is the conditional probability mass function for the transition from state $x_j$ to state $x_i$. We call this the one-step transition function. Normalisation requires that $\sum_i p(x_i|x_j) = 1$ for all $x_j \in \mathcal{X}$.

Note that in Eq. (17), we use capital letters to denote random variables and lower-case letters to denote specific values of these random variables. Probabilities are then denoted using the notation $\mathbb{P}(\cdot)$, while probability mass functions are denoted using $p(\cdot)$. We use this notation throughout the appendices.

Defining, $N = |\mathcal{X}|$ as the size of the state space, the transition matrix $\mathbf{P}$ is an $N \times N$ matrix with non-negative entries and column sums equal to one. The indices $i, j$ thus range from $1$ to $N$ inclusive.

The marginal probability distribution at time $t$ is the probability vector, $\mathbf{p}^{(t)}$, with components,

$$[\mathbf{p}^{(t)}]_i = \mathbb{P}(X(t) = x_i) = p^{(t)}(x_i)\,, \tag{18}$$

where $p^{(t)}(x_i)$ is the marginal probability mass function for state $x_i$ at time $t$. This gives the matrix equation for the evolution of the marginal distribution,

$$[\mathbf{p}^{(t+1)}]_i = \sum_j [\mathbf{P}]_{ij} [\mathbf{p}^{(t)}]_j = \sum_j p(x_i|x_j)\, p^{(t)}(x_j) = p^{(t+1)}(x_i)\,. \tag{19}$$

Such a Markov chain is thus a stochastic process parameterised by its transition matrix $\mathbf{P}$ and initial distribution $\mathbf{p}^{(0)}$. We denote this as,

$$\boldsymbol{X} \sim \mathrm{MC}(\mathbf{P}, \mathbf{p}^{(0)})\,. \tag{20}$$

Subject to technical conditions (of which Harris recurrence is sufficient (Brooks et al., 2011)), the Markov chain has a unique stationary distribution, $\boldsymbol{\pi}$, satisfying,

$$[\boldsymbol{\pi}]_i = \sum_j [\mathbf{P}]_{ij} [\boldsymbol{\pi}]_j = \sum_j p(x_i|x_j)\, \pi(x_j) = \pi(x_i)\,. \tag{21}$$

Under these conditions, for any initial probability vector, $\mathbf{p}^{(0)}$, and measurable function $f(x)$ defined on the state space $\mathcal{X}$, the time-averaged function,

$$\overline{f}_T = \frac{1}{T} \sum_{t=0}^{T-1} f(X(t)), \tag{22}$$

converges almost surely to the expectation of $f$ with respect to the stationary distribution, i.e.,

$$\overline{f}_T \xrightarrow[T\to\infty]{\text{A.S.}} \mathbb{E}_\pi[f(X)] = \sum_{i=1}^{N} f(x_i)\,\pi(x_i). \tag{23}$$

Furthermore, $\overline{f}_T$ converges in distribution as,

$$\sqrt{T}\left(\overline{f}_T - \mathbb{E}_\pi[f(X)]\right) \xrightarrow[T\to\infty]{D} \text{Normal}(0, \sigma_f^2), \tag{24}$$

where $\sigma_f^2$ is the asymptotic variance. This is defined as the limit of the scaled variance of the time-averaged function for the stationary process, $\boldsymbol{X}^* \sim \text{MC}(\mathbf{P}, \boldsymbol{\pi})$,

$$\sigma_f^2 := \lim_{T\to\infty} T\,\mathbb{V}[\overline{f}_T]. \tag{25}$$

Therefore, $\overline{f}_T$ provides a consistent and asymptotically normal estimator for the expectation of $f$ with respect to the stationary distribution, for any initial distribution.

## B.2. Markov Chain Monte Carlo

Markov Chain Monte Carlo (MCMC) and specifically the Metropolis-Hastings (MH) algorithm is a method to construct a transition matrix $\mathbf{P}$ such that the resulting Markov chain has some target stationary distribution, $\boldsymbol{\pi}$.

The Metropolis-Hastings one-step transition function can be defined as,

$$p_{\text{MH}}(x_i|x_j) = \begin{cases} q(x_i|x_j)\,\alpha(x_i|x_j) & x_i \neq x_j, \\ q(x_j|x_j)\,\alpha(x_j|x_j) + \sum_{z\neq x_j} q(z|x_j)\,[1 - \alpha(z|x_j)] & x_i = x_j. \end{cases} \tag{26}$$

Here, $q(x_i|x_j)$ is a proposal distribution and $\alpha(x_i|x_j)$ is the acceptance probability defined as,

$$\alpha(x_i|x_j) = \min\left(1, \frac{\pi(x_i)\,q(x_j|x_i)}{\pi(x_j)\,q(x_i|x_j)}\right), \tag{27}$$
$$= \min\left(1, r(x_i, x_j)\right),$$

where we have defined the Metropolis-Hastings ratio, $r(x_i, x_j)$.

A sufficient condition for a state to be the unique stationary distribution from some Markov chain is that it satisfies the detailed balance condition,

$$\pi(x_i)\,p(x_j|x_i) = \pi(x_j)\,p(x_i|x_j). \tag{28}$$

This can be used to confirm that the MH one-step transition function satisfies the detailed balance condition with respect to the target distribution.

Denote the transition matrix for the MH one-step transition function as $\mathbf{P}_{\text{MH}}^{(\mathbf{Q}, \boldsymbol{\pi})}$, where the superscript indicates that this transition matrix is constructed using the proposal distribution with transition matrix $\mathbf{Q}$ and target distribution with probability vector $\boldsymbol{\pi}$. It can be shown that the MH one-step transition function satisfies the detailed-balance condition with respect to the target distribution and, therefore, has $\boldsymbol{\pi}$ as its stationary distribution. The only remaining freedom is the choice of proposal distribution, $q(x_i|x_j)$.

Transition path sampling (TPS) is an MH method that makes a particular choice of proposal distribution suitable for cases where $x$ is an ordered sequence (i.e. a trajectory or sentence).

---

**Algorithm 1** Metropolis Hastings Algorithm

---

**Require:** Proposal Function $q$, Target Distribution $\pi$, Initial State $\mathbf{x}$, Number of TPS Steps $N$

$\quad n \leftarrow 0$

$\quad$ Create empty $N \times |\mathbf{x}|$ dimensional array $\bar{\mathbf{x}}$

$\quad$ **while** $n < N$ **do**

$\quad\quad$ Propose $\mathbf{x}' \sim q(\mathbf{X}|\mathbf{x})$

$\quad\quad$ Calculate acceptance probability, $\alpha = \min\left(1, \frac{\pi(\mathbf{x}')q(\mathbf{x}|\mathbf{x}')}{\pi(\mathbf{x})q(\mathbf{x}'|\mathbf{x})}\right)$

$\quad\quad$ Draw $u \sim \text{Uniform}(0, 1)$

$\quad\quad$ **if** $u \leq \alpha$ **then**

$\quad\quad\quad \mathbf{x} \leftarrow \mathbf{x}'$

$\quad\quad$ **end if**

$\quad\quad n \leftarrow n + 1$

$\quad\quad \bar{\mathbf{x}}_n \leftarrow \mathbf{x}$

$\quad$ **end while**

$\quad$ **Return:** $\bar{\mathbf{x}}$

---

In TPS, the proposal distribution randomly selects a position in the sequence and then changes only the elements of the sequence after this position according to some given function. Let $x_i = (x_i^{(1)}, x_i^{(2)}, \ldots, x_i^{(L_i)})$ denote a sequence of length $L_i$, and similarly for $x_j = (x_j^{(1)}, x_j^{(2)}, \ldots, x_j^{(L_j)})$ of length $L_j$. Denote a chosen cut position in the sequence as $c$ and use the notations, $x_{i/j}^{(\leq c)} = (x_{i/j}^{(1)}, x_{i/j}^{(2)}, \ldots, x_{i/j}^{(c)})$ and $x_{i/j}^{(>c)} = (x_{i/j}^{(c+1)}, x_{i/j}^{(c+2)}, \ldots, x_{i/j}^{(L_{i/j})})$ to denote the subsequences up to and including position $c$, and after position $c$ respectively. Therefore, $x_{i/j} = (x_{i/j}^{(\leq c)}, x_{i/j}^{(>c)})$ for any cut position $c$. The generic form of the TPS proposal distribution can then be written as,

$$q(x_i|x_j) = p_{\text{regen}}(x_i^{(>c)}|x_j^{(\leq c)})p_{\text{cut}}(c|x_j)\delta(x_i^{(\leq c)} - x_j^{(\leq c)}), \tag{29}$$

where $p_{\text{cut}}(c|x_j)$ is the distribution for selecting the cut position. The function $p_{\text{regen}}(x_i^{(>c)}|x_i^{(\leq c)})$ (re)generates the new subsequence after the cut position, and we have also chosen that this be conditional only on the subsequence up to and including the cut position, i.e., it is (conditionally) independent of the original subsequence after the cut position and with no explicit dependence on the cut position itself (it depends implicitly through the length of the prior subsequence),

$$p_{\text{regen}}(x_i^{(>c)}|x_j^{(\leq c)}) = p_{\text{regen}}(x_i^{(>c)}|x_j^{(\leq c)}, x_j^{(>c)}, c). \tag{30}$$

The delta function ensures that the subsequence up to and including the cut position remains unchanged.

The form of the MH ratio for this proposal distribution can then be written as,

$$r(x_i, x_j) = \frac{\pi(x_i)}{\pi(x_j)} \frac{p_{\text{regen}}(x_j^{(>c)}|x_i^{(\leq c)})}{p_{\text{regen}}(x_i^{(>c)}|x_j^{(\leq c)})} \frac{p_{\text{cut}}(c|x_i)}{p_{\text{cut}}(c|x_j)} \frac{\delta(x_i^{(\leq c)} - x_j^{(\leq c)})}{\delta(x_j^{(\leq c)} - x_i^{(\leq c)})}, \tag{31}$$

$$= \frac{\pi(x_i)}{\pi(x_j)} \frac{p_{\text{regen}}(x_j^{(>c)}|x_j^{(\leq c)})}{p_{\text{regen}}(x_i^{(>c)}|x_j^{(\leq c)})} \frac{p_{\text{cut}}(c|x_i)}{p_{\text{cut}}(c|x_j)} \delta(x_i^{(\leq c)} - x_j^{(\leq c)}), \tag{32}$$

where we have used the properties of the delta function in the second line to replace the conditioning subsequence in the numerator of the second fraction $x_i^{(\leq c)} \to x_j^{(\leq c)}$.

TPS takes the cut position to be uniformly distributed over the length of the sequence, i.e.,

$$p_{\text{cut}}(c|x_i) = \frac{1}{L_i} \quad c = 1, 2, \ldots, L_i, \tag{33}$$

$$p_{\text{cut}}(c|x_j) = \frac{1}{L_j} \quad c = 1, 2, \ldots, L_j. \tag{34}$$

Note that this choice means that the first element in the sequence cannot be changed, since the minimum $c = 1$ means that $x_{i/j}^{(\leq c)} = (x_{i/j}^{(1)})$ remains unchanged. The MH ratio is then,

$$r(x_i, x_j) = \frac{\pi(x_i)}{\pi(x_j)} \frac{p_{\text{regen}}(x_j^{(>c)}|x_j^{(\leq c)})}{p_{\text{regen}}(x_i^{(>c)}|x_j^{(\leq c)})} \frac{L_j}{L_i} \delta(x_i^{(\leq c)} - x_j^{(\leq c)}) . \tag{35}$$

Consider further the specific case that the target distribution is an exponentially reweighted distribution, i.e.,

$$\pi(x) = \frac{1}{Z} \tilde{\pi}(x) e^{-\lambda \phi(x)} , \tag{36}$$

$$= \frac{1}{Z} \tilde{\pi}(x^{(\leq c)}) \tilde{\pi}(x^{(>c)}|x^{(\leq c)}) e^{-\lambda \phi(x)} , \tag{37}$$

for any cut position $c$ and where $\tilde{\pi}(x)$ is some base distribution. The ratio of probabilities in the target distribution is then,

$$\frac{\pi(x_i)}{\pi(x_j)} = \frac{\tilde{\pi}(x_i)}{\tilde{\pi}(x_j)} e^{-\lambda(\phi(x_i) - \phi(x_j))} , \tag{38}$$

$$= \frac{\tilde{\pi}(x_i^{(\leq c)}) \tilde{\pi}(x_i^{(>c)}|x_i^{(\leq c)})}{\tilde{\pi}(x_j^{(\leq c)}) \tilde{\pi}(x_j^{(>c)}|x_j^{(\leq c)})} e^{-\lambda(\phi(x_i) - \phi(x_j))} . \tag{39}$$

This gives the MH ratio,

$$r(x_i, x_j) = \frac{\tilde{\pi}(x_i^{(\leq c)}) \tilde{\pi}(x_i^{(>c)}|x_i^{(\leq c)})}{\tilde{\pi}(x_j^{(\leq c)}) \tilde{\pi}(x_j^{(>c)}|x_j^{(\leq c)})} e^{-\lambda(\phi(x_i) - \phi(x_j))} \frac{L_j}{L_i} \delta(x_i^{(\leq c)} - x_j^{(\leq c)}) \frac{p_{\text{regen}}(x_j^{(>c)}|x_j^{(\leq c)})}{p_{\text{regen}}(x_i^{(>c)}|x_j^{(\leq c)})} , \tag{40}$$

$$= \frac{\tilde{\pi}(x_i^{(>c)}|x_j^{(\leq c)})}{\tilde{\pi}(x_j^{(>c)}|x_j^{(\leq c)})} e^{-\lambda(\phi(x_i) - \phi(x_j))} \frac{L_j}{L_i} \delta(x_i^{(\leq c)} - x_j^{(\leq c)}) \frac{p_{\text{regen}}(x_j^{(>c)}|x_j^{(\leq c)})}{p_{\text{regen}}(x_i^{(>c)}|x_j^{(\leq c)})} , \tag{41}$$

where we have again used the properties of the delta function in the second line. This expression holds for any choice of regeneration function, $p_{\text{regen}}$ (under the conditional independence assumptions outlined above).

A natural choice is to define $p_{\text{regen}}$ using the base distribution, $\tilde{\pi}(x)$, i.e.,

$$p_{\text{regen}}(x_i^{(>c)}|x_j^{(\leq c)}) = \tilde{\pi}(x_i^{(>c)}|x_j^{(\leq c)}) . \tag{42}$$

This gives the MH ratio,

$$r(x_i, x_j) = \frac{\tilde{\pi}(x_i^{(>c)}|x_j^{(\leq c)})}{\tilde{\pi}(x_j^{(>c)}|x_j^{(\leq c)})} e^{-\lambda(\phi(x_i) - \phi(x_j))} \frac{L_j}{L_i} \delta(x_i^{(\leq c)} - x_j^{(\leq c)}) \frac{\tilde{\pi}(x_j^{(>c)}|x_j^{(\leq c)})}{\tilde{\pi}(x_i^{(>c)}|x_j^{(\leq c)})} , \tag{43}$$

$$= e^{-\lambda(\phi(x_i) - \phi(x_j))} \frac{L_j}{L_i} \delta(x_i^{(\leq c)} - x_j^{(\leq c)}) , \tag{44}$$

where we have again used the properties of the delta function in the third line, allowing all terms involving $\tilde{\pi}$ to cancel.

With regard to the implementation of MCMC, we provide two algorithms: Algorithm 1 details the general Metropolis-Hastings algorithm, whilst Algorithm 2 gives a pseudocode implementation of the exact algorithm (based on TPS) used for sampling the results in Figure 2 of the main text. Note that in Algorithm 2, we use the definition $\mathbf{X}_{n:T} = \emptyset$ when $n > T$. Many more details concerning MCMC and MH can be found in (Brooks et al., 2011) and references therein.

## C. Implementation Details, Tips and Tricks for Rare Event Sampling in Language Models

### C.1. Annealed TPS

While any initial state can be used for MCMC, the chain must (at least) be run for a sufficient number of steps to reach the steady-state distribution, and thus begin obtaining samples from the target distribution. As such, for some fixed number of

---

**Algorithm 2** Annealed Transition Path Sampling for Language Modelling

---

**Require:** Language Model $\mathbb{P}_{\mathcal{M}}$, Prompt $\mathbf{x}_{1:t}$, Observable $\phi$, Biases $(\lambda_1, \ldots, \lambda_K)$, Steps per Bias $N$, Trajectory Length $T$
    Sample initial completion, $\mathbf{x}_{t+1:T} \sim \mathbb{P}_{\mathcal{M}}(\mathbf{X}_{t+1:T}|\mathbf{x}_{1:t})$
    Calculate initial observable value, $o \leftarrow \phi(\mathbf{x}_{1:t}, \mathbf{x}_{t+1:T})$
    Create $K \times N$ matrix $\Phi$ filled with 0
    **for** $k \in (1, \ldots, K)$ **do**
        $n \leftarrow 0$
        **while** $n < N$ **do**
            Choose a random integer $\tau$ uniformly from $1, \ldots, T$
            Sample the new proposal, $\mathbf{x}'_{t+\tau+1:T} \sim \mathbb{P}_{\mathcal{M}}(\mathbf{X}_{t+\tau+1:T}|\mathbf{x}_{1:t}, \mathbf{x}_{t+1:t+\tau})$
            Calculate proposal observable value, $o' = \phi(\mathbf{x}_{1:t}, \mathbf{x}'_{t+1:T})$, where $\mathbf{x}'_{t+1:T} = (\mathbf{x}_{t+1:t+\tau}, \mathbf{x}'_{t+\tau+1:T})$
            Calculate acceptance probability, $\alpha = \min\left[1, \exp(-\lambda_k(o' - o))\right]$
            Draw $u \sim \text{Uniform}(0, 1)$
            **if** $u \leq \alpha$ **then**
                $\mathbf{x} \leftarrow \mathbf{x}', o \leftarrow o'$
            **end if**
            $\Phi_{k,n} \leftarrow o$
            $n \leftarrow n + 1$
        **end while**
    **end for**
    **Return:** $\Phi$

---

steps/computational budget, choosing an initial state more likely in the target steady-state distribution will result in samples with a distribution more representative of the target distribution.

Annealed TPS aims to start with an initial state more representative of the steady-state for any given bias, by progressively increasing the bias over time. Since the steady-state distributions for nearby values of the bias are similar to each other, the final state from sampling with some bias $\lambda_k$ should be more representative of the steady-state distribution for bias $\lambda_{k+1}$ than a random initial state when $\lambda_k - \lambda_{k+1}$ is small.

The trade-offs for this are two-fold: firstly, due to the sequential approach, sampling for each bias cannot be done in parallel with this basic annealing approach; secondly, the samples from each bias will not be independent (due to the annealing) due to weak correlations along the shared MCMC chain.

If we wish to sample from a single biased distribution, annealing can still be beneficial when the target distribution is very different from the base distribution, as the smooth change in distribution can lead to better mixing. In that case, the annealed samples for non-target distributions (i.e. those with intermediate biases) can then be discarded as burn-in.

For applications involving MBAR, or related methods, we instead wish to sample from multiple biased distributions, and require sufficient overlap between distributions (see Section C.8).

### C.2. Burn-in

Burn-in is the process of discarding some initial set of samples from MCMC (Roy, 2020). This is motivated by the fact that the initial distribution might be far from the target stationary distribution, so that the samples obtained in the initial steps of MCMC might not be representative of the target distribution. In this sense, burn-in can be seen as a way of establishing a good starting point for MCMC (Brooks et al., 2011).

### C.3. Gelman-Rubin Statistic

The Gelman-Rubin Statistic is a measure of the convergence of MCMC (Roy, 2020), i.e. the degree to which multiple chains have mixed and converged to the same distribution. For each bias $\lambda$, we run $J$ parallel chains for $L$ steps (excluding burn-in), to obtain samples $(x_1^{(j)}, \ldots, x_L^{(j)})$ for chain $j$, and thus the values of our observable $(y_1^{(j)}, \ldots, y_L^{(j)})$, where $y_i^{(j)} = f(x_i^{(j)})$.

We can then calculate the GR statistic as,

$$\text{GR} = \frac{\frac{L-1}{L}W + \frac{1}{L}B}{W},$$ (45)

where $B$ and $W$ are defined as,

$$B = \frac{L}{j-1}\sum_{j=1}^{J}(\bar{y}_j - \bar{y}_*)^2, \qquad W = \frac{1}{J}\sum_{j=1}^{J}\left(y_i^{(j)} - \bar{y}_j\right)^2,$$ (46)

where $\bar{y}_j$ is the mean of chain $j$ and $\bar{y}_*$ is the mean of the means of the chains,

$$\bar{y}_j = \frac{1}{L}\sum_{i=1}^{L} y_i^{(j)}, \qquad \bar{y}_* = \frac{1}{J}\sum_{j=1}^{J}\bar{y}_j.$$ (47)

This statistic tends towards 1 as $L$ goes to infinity and the mean of the variances of the chains, $B$, goes to 0. We accept samples from biases that have a Gelman-Rubin statistic of less than 1.1.

An adaptive scheme to determine when to end sampling for a given distribution could be implemented by calculating the GR statistic at intervals during sampling and stopping once the statistic is below some threshold.

### C.4. Choosing an Observable

Choosing a suitable observable, $\phi(x)$, to bias towards is essential for the rare event sampling methodologies presented in the main text to be effective.

Consider then the case that we wish to evaluate the expectation of some observable $A(x)$. While a natural choice for the biasing observable is to use $\phi(x) = A(x)$, this is not strictly necessary and may not be optimal. Instead, one can choose another observable $\phi(x) \neq A(x)$ that is expected to correlate with $A(x)$, but has more favourable properties. This could include the fact that $\phi(x)$ is smoother, or cheaper to calculate than $A(x)$.

This is particularly important when $A(x)$ takes on only a small number of possible values, e.g. a binary observable, because biasing directly on $A(x)$ will not be effective. MCMC works by gradually shifting the distribution in the direction specified by the bias and, as such, these techniques will not be effective when applied to observables with only a small number of possible values (such as binary observables) because the distribution cannot be shifted gradually. One should thus bias with continuous observables, or at least observables with a large number of possible values.

Taking AI safety as an example, often we wish to mark completions as 'safe' and 'unsafe'. If a model has been sufficiently finetuned, it is likely that unsafe generations are rare enough that direct sampling is infeasible. Since biasing towards this binary observable is not effective, we could instead bias towards some proxy, such as the count of some set of words that suggest the text may be unsafe.

### C.5. Choosing Biases

Choosing the right set of biases $\{\lambda_k\}_{k=1}^{K}$ is also essential to ensure the usefulness of this methodology. If biases are too large, it may take too long for TPS to converge. Furthermore, as we often found, larger biases can become stuck at a single sample with some extreme value of the observable. In this case, it is very unlikely for TPS to leave this state, since any proposed change will likely be less extreme and therefore have a low probability of acceptance. On the other hand, if biases are too small, the slight increase in coverage of the tails from the biased distribution will not be enough to outweigh the effect of the correlated samples.

We found that if we expect the highest/lowest values of the observable to be on the order of $10^m$, biasing up to around $\pm 10^{-m}$ seemed to provide good coverage of rare events without excessive correlation caused by low acceptance rates. However, in general this will depend on all factors of the problem, including the model, prompt, observable and algorithmic details. As such, experimentation is likely necessary on a case-by-case basis to find the best set of biases.

The density of biases used is also important to ensure distributional overlap between adjacent biases, as discussed in Section C.8. In systems exhibiting phase transitions, it may be necessary to use a denser set of biases in the region of the phase transition to ensure sufficient overlap, as such regions correspond to drastic changes in distributional properties.

An adaptive bias selection scheme could be used to satisfy a combination of the desired range of observable values to cover, and the desired overlap between distributions.

## C.6. Choosing a Proposal Function

In this work, we consider a proposal function that is proportional to the unbiased dynamics of the original model, i.e.,

$$q(\mathbf{X}_{t:T}|\mathbf{x}_{1:t}) \propto \mathbb{P}_{\mathcal{M}}(\mathbf{X}_{t:T}|\mathbf{x}_{1:t}). \tag{48}$$

In the context of TPS, this is used as the regeneration function, $p_{\text{regen}}(x_i^{(>c)}|x_j^{(\leq c)}) = \mathbb{P}_{\mathcal{M}}(x_i^{(>c)}|x_j^{(\leq c)})$, which has the advantage that the acceptance probability becomes independent of the base distribution, see (44).

However, making a different choice is also possible and could dramatically decrease the convergence time by increasing the probability of accepting the proposed changes, allowing a reduction in samples discarding to burn-in time. Further, it could also reduce correlation between samples, resulting in a greater effective sample size for estimates.

Here we list examples that could be explored in future works. In each case we use the notation of (48), with the understanding that this can be used as the regeneration function in TPS. In all examples, one can always calculate the acceptance probability from the definition in (27). Note that these examples are not necessarily mutually exclusive, and combinations can be considered.

### C.6.1. PROMPT ENGINEERING AND JAILBREAKING

One simple way to change the proposal function is to continue to use the same model, but changing the prompt. For example, if we are trying to sample the distribution $\mathbb{P}_{\mathcal{M}}(\mathbf{X}_{t+1:T}|\mathbf{x}_{1:t})$ biased towards some observable $\phi(x)$, we could use,

$$q(\mathbf{X}_{t:T}|\mathbf{x}_{1:t}) \propto \mathbb{P}_{\mathcal{M}}(\mathbf{X}_{t:T}|J(\mathbf{x}_{1:t})), \tag{49}$$

where $J$ is some modification to the prompt that makes extreme values of $\phi(x)$ more likely. In the case of AI safety, these modifications $J$ would take the form of jailbreaking or red teaming prompts.

### C.6.2. SMALLER MODELS

If we are trying to target rare events in a large model, the number of samples required could be too expensive, even with the benefits offered through TPS and MBAR. However, the computational cost could be greatly reduced by choosing a smaller model as a proposal function. That is,

$$q(\mathbf{X}_{t:T}|\mathbf{x}_{1:t}) \propto \mathbb{P}_{\mathcal{M}_{\text{small}}}(\mathbf{X}_{t:T}|\mathbf{x}_{1:t}). \tag{50}$$

Then, to generate $N$ tokens, instead of needing $N$ calls to the larger model, we would need $N$ calls to the smaller model, plus just a single call to the larger model (which is needed to compute the probability of the completion under the target dynamics in the acceptance probability). This may increase the number of steps needed to reach convergence, since the distributions of the model will be slightly different, however it could significantly decrease the computation required due to the decreased number of calls to the expensive model. For this to be effective, the large and small model will have to have sufficiently 'similar' distributions.

### C.6.3. FINETUNING

Reinforcement Learning could be used to finetune a model to produce extreme values of the observable of interest. This finetuned model could then be used as a proposal to evaluate rare events of other models, with respect to this observable. That is,

$$q(\mathbf{X}_{t:T}|\mathbf{x}_{1:t}) \propto \mathbb{P}_{\mathcal{M}(\theta^*)}(\mathbf{X}_{t:T}|\mathbf{x}_{1:t}), \tag{51}$$

where $\mathcal{M}(\theta^*)$ is the model with fitted parameters $\theta^*$ after finetuning.

Whilst the initial finetuning process could be computationally expensive, it could significantly decrease the number of samples required to produce a good estimate of the tails in the model of interest. Furthermore, since the model can be

finetuned over multiple prompts, it could be an effective way to calculate tail distributions over an entire dataset. By finetuning towards the exponential distribution directly, the proposal function would be more likely to propose samples representative of the target distribution, increasing acceptance rates and decreasing correlation between samples.

We note that, in principle, a model perfectly fitted to the exponentially reweighted distribution would then mean that uncorrelated samples could be generated directly from the finetuned model, without a need for MCMC at all. However, most RL techniques used for finetuning LLMs do not guarantee convergence to the global minimum distribution, while MCMC guarantees sampling from this in the limit of infinite steps. Nonetheless, this would be a key area for future research.

### C.6.4. Infilling

The version of TPS applied in this work and detailed in Algorithm 2 is sometimes called forward shooting due to the replacement of the future after the cut. One drawback of this approach is that changes to the beginning of trajectories are relatively unlikely. There are two contributing factors to this. Firstly, if we are proposing an edit to a completion with $N$ tokens, there is an $n/N$ chance that we choose to cut at a point that modifies the $n$th token, meaning the probability is very low if $n$ is small relative to $N$. Secondly, even if a proposal is made that changes an early token, nearly all of the trajectory that is contributing to the extreme value of the observable is discarded, meaning the proposal will likely be from the bulk of the original distribution. Since the observable value for this proposal is likely to be much less extreme, the probability of acceptance will be very low. As such, the generated samples are likely to be very correlated, particularly towards the beginning of the completions.

In contrast to the models considered in many other applications of TPS such as statistical mechanics, there are in fact LLMs capable of filling in the middle of trajectories, where context is provided both before and after the completion. These are known as infilling models. Such an infilling model could potentially be used as a proposal function, eliminating this shortcoming of forward shooting TPS.

### C.7. Parallel Tempering

Parallel tempering is an extension to MCMC, used to decrease the autocorrelation time within each chain (Earl & Deem, 2005). In parallel tempering, several MCMC chains at varying temperatures are run in parallel. Then, at each MCMC step, there is a probability of exchanging the current states between the different chains. This means, for example, there is some probability that a chain with a strong negative bias might be in a state typical of a distribution with a strong positive bias.

To satisfy detailed balance, the probability of proposing a swap between any two states $i$ and $j$ is equal for any $i, j$ and given any history of the chains. Then, a proposed swap between the states in chains $i$ and $j$ is accepted with probability,

$$A(x_j, x_i | x_i, x_j) = \min\left(1, e^{(\lambda_i - \lambda_j)(\phi(x_i) - \phi(x_j))}\right). \tag{52}$$

Parallel tempering works to prevent chains at high biases from getting stuck by giving them some possibility of moving to a much less biased state, thus breaking out of a local minimum. In our experiments, we found that chains biased strongly towards the ARI score would often get stuck in highly repetitive states, and be unable to escape. Hence, parallel tempering could offer a good solution to prevent this.

### C.8. Distribution Overlap

For MBAR to be accurate, overlaps between distributions must be sufficient such that a matrix of overlaps can not be reordered to have disconnected blocks (Klimovich et al., 2015). This can be represented using the overlap matrix,

$$O_{ij} = \mathbb{E}_{X \sim \mathbb{P}_j}\left[\frac{N_i p_i(X)}{\sum_{k=1}^{K} N_k p_k(\mathbf{X})}\right], \tag{53}$$

where $N_1, \ldots, N_K$ are the number of samples taken from distribution $p_1, \ldots, p_K$. Each matrix element $O_{ij}$ is the probability of observing a sample from distribution $i$ in distribution $j$. We calculate this using a Monte-Carlo estimate with the samples from the mixture distribution as,

$$\tilde{O}_{ij} = \sum_{n=1}^{N}\left[\frac{N_i p_i(x_n)}{\sum_{k=1}^{K} N_k p_k(x_n)} \cdot \frac{p_j(x_n)}{\sum_{k'=1}^{K} N_{k'} p_{k'}(x_n)}\right]. \tag{54}$$

If the overlap between neighbouring states is larger, the MBAR estimate should be more accurate. Klimovich et al. suggest that values as low as 0.03 can be used to yield reliable estimates, and as such we use this as a cutoff value.

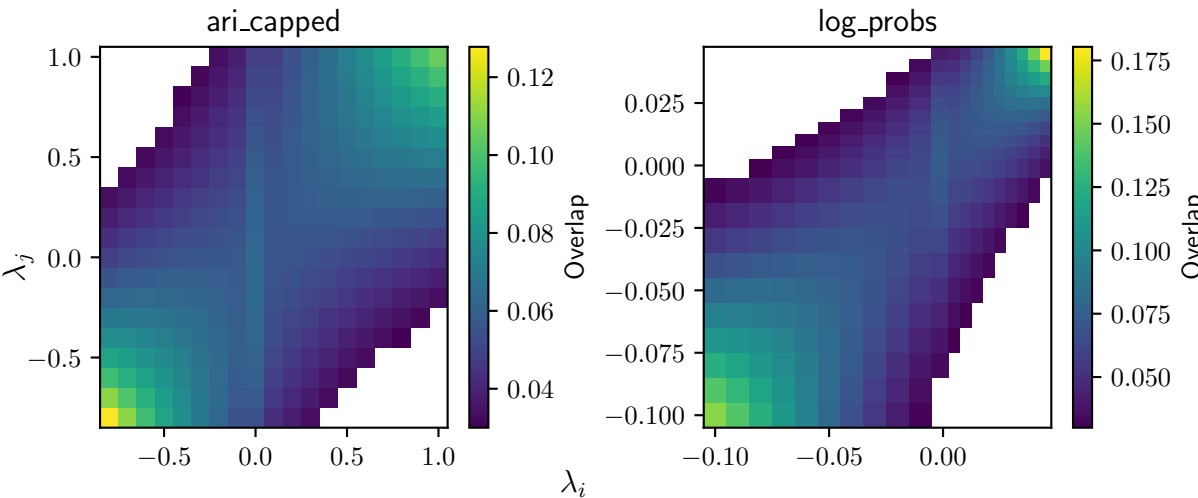

*Figure 6.* **Overlap of Biased Distributions for ARI and Logprobs.**. Estimates for the distributional overlap of the distributions biased towards ARI and the Log Probability, calculated using the data from Figure 2. The heatmap displays an estimate for the probability of observing a sample from distribution $i$ in distribution $j$. Distributions with overlap of less than 0.03 are shown in white.

It can be seen from the off-diagonal elements of Figure 6 that there is sufficient overlap between the neighbouring distributions for MBAR analysis to be valid. Furthermore, the overlap between the biased distributions could be used to guide additional sampling. If somewhere in the annealing schedule has low overlap, additional TPS chains could be run to gather samples to fill in the grid of biases between these regions, preventing the need to rerun all of the trajectories.

Note that when $N_i = N_j$, $O_{ij}$ is symmetric. Since the unbiased distribution, $\lambda = 0$, has a different number of samples to the biased distributions, this symmetry is broken, which can be seen by the $\lambda_i = 0$ column in Figure 6.

## D. Comparison of Histogram Estimations for Different Starting Prompts

To show that the methodology described in Sect. 3 is applicable across a range of prompts, Figure 7 displays reweighted and unbiased histograms for prompts with a range of ARI values. In nearly all cases, the biased histogram provides estimates further into the tails than the unbiased, with the exception of the left tail in Figure 7(c). This suggests that a different annealing schedule for this prompt could be beneficial.

When applying these techniques to a large number of prompts, it will be infeasible to manually decide on an annealing schedule for each prompt. As such, adaptive ways of switching the bias, such as switching biases only when the Gelman-Rubin statistic is below a certain threshold, could be a potential avenue of future research.

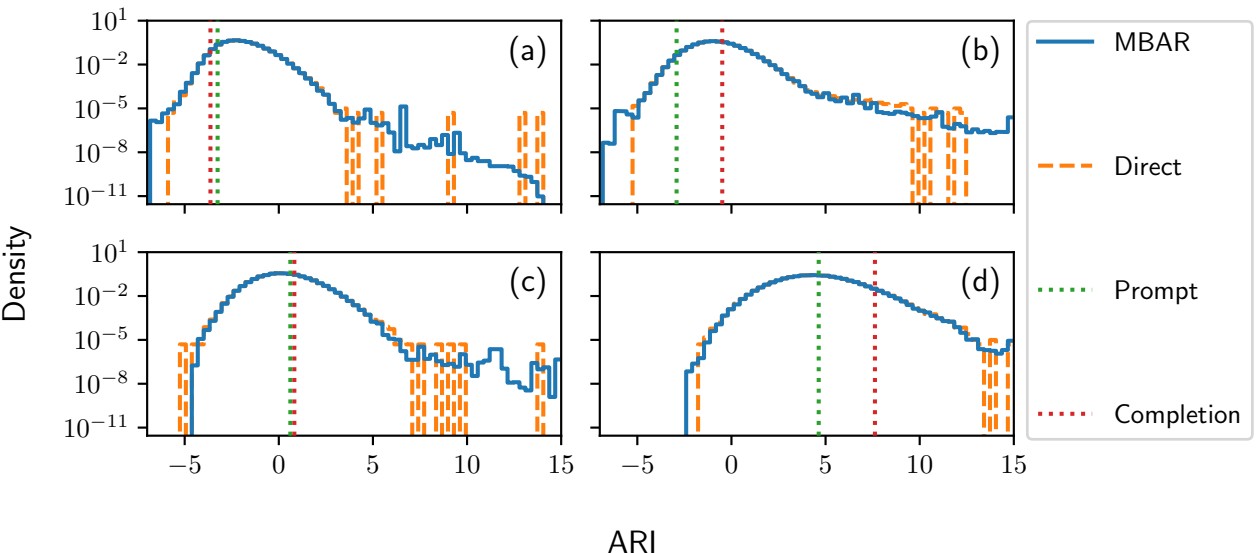

*Figure 7.* **Reweighted Histograms for Various Starting Prompts.** Histograms made through TPS and MBAR (blue) and Direct Sampling (orange, dashed), for various starting prompts. The ARI for the prompt and reference completion are shown by the green and red dotted lines. The prompts are taken from the subset of datapoints with more than 272 tokens in the the validation split of the TinyStories dataset. Figures (a), (b), (c) and (d) correspond to the datapoints at the 0%, 33%, 67% and 100% percentiles of this subset, when ordered by ARI scores. The ARI scores of the corresponding prompts (green) and completions (red) taken from these datapoints shown by the horizontal dashed lines. The biased histograms are made using the same biases as Figure 2, with a reduced steps per bias of 20, 000.

