# OpenReview forum: "Rare Event Analysis of Large Language Models"
_ICML.cc/2026/Conference — ICML 2026 spotlight_

### Official Review · Reviewer_ZdB6 · 2026-03-11

**Soundness:** 2
**Presentation:** 3
**Significance:** 1
**Originality:** 2
**Overall Recommendation:** 2
**Confidence:** 4

**Summary:**

The paper presents a complete framework to analyze rare events in LLMs. Practical implementation spanning theory, generation strategies, probability estimation and error analysis are provided in the paper.

**Compliance With Llm Reviewing Policy:**

Affirmed.

**Key Questions For Authors:**

1. In the methodology section, only a sampling strategy and a single definition are provided, while the framework and analysis approach are not presented. I want to know what specific framework and methodology the article proposes.

2. The paper just evaluate on a small LLM, although their the results show that the computational costs could be extended to larger LLM, other performance remains uncertain. Hence, the conclusion does not achieve the expected goals which are stated in "Aims and Objectives".

3. Comparative studies with SOTA are expected, so that the generation effect, correctness of estimation can be proved efficient for the framework.

**Limitations:**

yes

**Strengths And Weaknesses:**

-The framework and methodology are not provided, rendering it incomplete.
-Experiments were only conducted on small models.
-Necessary comparative studies have not been carried out.

---

> ### Author Rebuttal · Authors · 2026-03-31
>
> We thank the Reviewer for their report. While we regret their overall assessment, we hope that it is in part due to misunderstanding that we aim to clarify in our responses below.
>
> ***(viii) Comment by Reviewer: “The framework and methodology are not provided, rendering it incomplete […] only a sampling strategy and a single definition are provided […] I want to know what specific framework and methodology the article proposes.”***
>
> **Response**: Our paper indeed presents a complete mathematical framework for describing rare events and a methodology for quantifying them, and therefore we are puzzled by this remark.
>
> A general methodology for rare events is given in Sec. 3, based on exponential tilting of reweighted trajectories, to define generating functions of the observables of interest and to systematically bias towards rare occurrences. The methodology is based on importance sampling, histogram reweighting and transition path sampling, thus adapting SOTA techniques from statistical mechanics to the LLM context. Details specific to the computational experiments are provided in Sec. 4, which compares these efficient methods to direct sampling. The appendices give pseudocode for the algorithms. The SM includes working code that generates the same figures as presented in the main text.
>
> We clarify that we understand this as “conceptual” or “mathematical” framework, rather than “machine learning framework” (PyTorch, etc). While numerical experiments require the latter for implementation (see SM), the overall approach does not depend on this. Specifically, by a framework for rare event analysis for LLMs we mean what should be computed (desired outcomes) and how to do it (generic methods).
>
> In our framework: (a) desired outcomes are the probabilities of occurrence and the characteristic properties of rare outputs; and (b) the way to do it is by adapting SOTA methods from statistical mechanics designed for this purpose (importance sampling-based methods for probability estimation, EDA/unsupervised learning techniques for characterisation).
>
> A key aim was to bring this large body of literature (from statistical mechanics, as applied in physics and chemistry) to the attention of a machine learning audience, namely those with expertise in language models, so that rare event analysis for language models could be more rapidly developed.
>
> Our framework is described throughout the paper. However, to address the Reviewer’s comment we will include a more explicit summary in the introduction to aid clarity. Succinctly, a revised introduction will include:
> - “Setup”, that a REA problem requires a probability distribution generating outcomes (here a LLM) and an observable of interest (readability, safety, etc.), as described in Sec. 3.1, with specific examples in Sec. 4.
> - “Aim”, that understanding rare events requires estimating their probabilities and exploring their properties.
> - “Estimation”,  explaining the large literature of methods from statistical mechanics, including exponential tilting, histogram reweighting, and transition path sampling, describing this theoretical framework in Sec. 3 and demonstrating its use for LLMs in Sec. 5.
> - “Exploration”, outlining how to understand and control rare model outputs one must study their properties in a domain and problem dependent way; we illustrate it with the case of suppressing rare high-ARI completions (i.e. complex text as undesirable); we show that even supposedly well-aligned models will have rare events that depart from alignment, which with our techniques can be quantified and studied.
>
> ***(ix) Comment: “Experiments were only conducted on small models [… extended to larger LLMs] performance remains uncertain […] does not achieve the expected goals which are stated in Aims and Objectives.”***
>
> **Response**: We fully understand the concern of the Reviewer here regarding small models. See our response to Reviewer pVSq’s comment (ii) for further details.
>
> While we agree that demonstrations on larger models are desirable, we disagree that we have not achieved the goals stated in “Aims and Objectives”. We show substantial efficiency gains over direct sampling for small models, and the large body of research on importance sampling strongly suggests these efficiency gains would hold for larger models. We further argue in Sec. 4 that the token costs required are feasible for a large tech lab. We also expect future work to lead to substantial efficiency improvements, as  outlined in the outlook and SM, making the approach accessible to many research projects including for larger models.
>
> ***(x) Comment: “Comparative studies with SOTA are expected […]”***
>
> **Response**: We provide a comprehensive answer to this point in our response to Reviewer LyuJ’s comment (vii). In short, to our knowledge, the SOTA for sampling rare events in LLMs is direct sampling, which we compare to and show the large efficiency gains of using statistical mechanics methods.

---

### Official Review · Reviewer_LyuJ · 2026-03-13

**Soundness:** 3
**Presentation:** 3
**Significance:** 3
**Originality:** 4
**Overall Recommendation:** 5
**Confidence:** 4

**Summary:**

Thank you for the opportunity to review the paper. I like your approach to REA applied to LLMs. I especially appreciate the framework that includes methods for generating rare samples, estimating their probabilities, computing confidence intervals, and exploring the structure of these events. The authors address a significant challenge of how to estimate and explore extremely low-probability behaviors that may become relevant at deployment scale.

**Compliance With Llm Reviewing Policy:**

Affirmed.

**Final Justification:**

I had only a few concerns during the review, which authors have amply and effectively addressed. Together with my original evaluation of the paper novelty, my overall recommendation is "accept."

**Key Questions For Authors:**

None

**Limitations:**

Yes

**Strengths And Weaknesses:**

Strengths: The paper is very well written and easy to follow. The conceptual framing is novel, borrowing from statistical physics and computational chemistry. Well-developed pipeline for rare-event analysis. This comprehensive approach is valuable to researchers studying rare LLM behaviors. Experimental demonstration is clear and the qualitative analysis is quite insightful.

Limitations: the scale of experiments is limited, the set of observables is narrow, and the computational cost is high. I also thought your comparison with alternatives is somewhat limited as well.

EDIT: Adjusting review score following the author rebuttal; all my concerns were addressed.

---

> ### Author Rebuttal · Authors · 2026-03-31
>
> We thank the Reviewer for their overall assessment. The summary and strengths sections indicate a careful reading of the paper for which we are very grateful.
>
> ***(v) Comment by Reviewer: “[…] the scale of experiments is limited, the set of observables is narrow […]”***
>
> **Response**: We agree with the Reviewer that the set of observables studied explicitly is limited. It is clearly desirable for future work to include the analysis of a wider range of observables in a wider range of scenarios. However, the general methodology that we present here remains the same, and in this paper by focusing on simple (and well-established) metrics like the ARI and Log-Probabilities we can clearly demonstrate how this framework is applied to study the rare events in a specific problem of interest. Using these two observables as concrete examples allows us to demonstrate in detail how to study the properties of their rare events, as is done in Sec. 6. With the general framework in place, it is the possible to move to other domains, with AI safety a very natural and important direction. The methodology in such an application would be the same one presented here but using domain-specific metrics, and therefore specialised knowledge. We chose here to illustrate the general framework with a use case accessible to a broad audience, and to leave specialised applications of our approach to future work.
>
> ***(vi) Comment: “[…]  and the computational cost is high.”***
>
> **Response**: We agree that the computational cost is high relative to what one might expect from experience with direct sampling (or any similar local decoding strategy). However, for the estimation of rare event probabilities (or, equivalently, the generation of rare completions) the method we present here is orders of magnitude computationally more efficient than direct sampling (which is the current standard approach for this problem).
>
> Additionally, as we argue in Sec. 4, the total computational cost of the method as given is in fact feasible for a large tech company applying this methodology to a large model, which would not be the case if using brute force direct sampling.
>
> Our aim here is to provide a practical starting point for studying rare events in LLMs. We naturally hope that future work will allow for significant further efficiency improvements to lower the resource requirements. To this end, we provide examples of potential research directions that could improve the computational efficiency of this framework in Sec. 7 and Appendix C.
>
> ***(vii) Comment: “I also thought your comparison with alternatives is somewhat limited as well.”***
>
> **Response**: We completely agree with the Reviewer that it is highly desirable to perform large scale comparisons with other methods (including SOTA), and that this is an important validation step for any proposed methodology. However, to the best of our knowledge, for LLMs there are no methods for rare event analysis, and in particular for the estimation of their probabilities, beyond direct sampling.
>
> For example, while we mention a number of potential comparable approaches in Sec. 2, none of these other methods have yet been fully developed for the goal of rare-event probability estimation for longer generations as presented for example in the histograms of Fig. 3. We note that we do refer specifically to a couple of closely related works concerning rare event estimation, but these do not directly address the same question.
>
> As such, the only method that can be directly compared is direct sampling, which is thus the SOTA to contrast against. For this case we do provide extensive comparisons, demonstrating dramatic improvements in efficiency with our framework.
>
> Given that the SOTA is brute force sampling, the aim of our paper is to provide a general framework and an initial study. Our work thus serves as both a guide and as a baseline for further comparison, for example by building on the methods we outline in our response to (vi) above.  The development of alternative methods and comprehensive comparative analysis is indeed a major avenue for future work.
>
> On revision, we will add a short paragraph to the introduction to emphasise and clarify this point.

---

> > ### Author Rebuttal · Reviewer_LyuJ · 2026-03-31
> >
> > I already liked the paper in its original form, because I appreciate the novelty of conceptual framing. Author responses further assured me that I was correct, and their paper represents good, solid work. I am fully satisfied with the response and adjusting my score to "accept."

---

> > > ### Author Response · Authors · 2026-04-08
> > >
> > > Thank you again for your careful reading of our paper. We hope that the changes inspired by your comments will make this paper accessible to a wider audience.

---

### Official Review · Reviewer_pVSq · 2026-03-18

**Soundness:** 3
**Presentation:** 3
**Significance:** 3
**Originality:** 4
**Overall Recommendation:** 5
**Confidence:** 3

**Summary:**

Authors present an end-to-end process to analyze rare events in LLMs. First they present the building blocks to produce rare tokens via exponentially reweighted token distributions. Next they discuss evaluation tools automated readability index (ARI) as an example evaluation metric alternative to log probability.

**Compliance With Llm Reviewing Policy:**

Affirmed.

**Final Justification:**

I believe my concerns have been adequately addressed.

**Key Questions For Authors:**

1. You use ARI which has a range of 1 to 14 that corresponds to reader age. Figure 3a and 3b show ARI range of -5 to 15. How would you interpret ARI scores outside the human-corresponding age?

**Limitations:**

Yes

**Strengths And Weaknesses:**

Strengths
1. Well motivate work. Rare event analysis for language models seem disconnected, this paper attempts to offer a cohesive idea that include how to induce rare event trajectories and how to analyze them.
2. The paper provides experimentations to show how the pipeline works by using TinyStories. From Analysis and estimation for rare event probabilities to sampling of rare event trajectories.

Weakness
1. This paper interestingly describes a method to produce rare-event tokens/trajectories but most of the paper focuses on proving the validity of the approach through experimentations. This is no doubt commendable, but what is missing is how to utilize this is areas or domains the target reader may be interested. Using Tiny Stories is a valid choice and the use of ARI as a metric is well justified, but if the method is indeed cost efficient, extending analysis to larger models with pretraining sets that are also public such as OLMO or Pythia should also be done to be more convincing.
2. I think the cases which the authors describe are weak cases in which the proposed framework would be useful. For example, they highlight that analysis from this framework could be used to guide development and reveal failure modes which could then be filtered by removing high-ARI trajectories. However, repetition penalty is a widely used method that is even accessible in libraries such as VLLM and SGLang.

---

> ### Author Rebuttal · Authors · 2026-03-31
>
> We thank the Reviewer for their insightful assessment, indicating a careful reading of the paper. We hope the Reviewer will be satisfied with our responses. We have anticipated several potential additions to the paper based on the Reviewer’s comments that we believe will enhance the clarity of the text.
>
> For all Reviewers, we quote each specific Reviewer comment followed by our response.
>
> ***(i) Comment by Reviewer: “[…] what is missing is how to utilize this is areas or domains the target reader may be interested.”***
>
> **Response**: We agree it would be helpful for readers to address more explicitly how to adapt our methods to their domain. As presented, the framework/methodology is generic and domain-specific questions are targeted by choosing two observables: the “target observable” for which estimates (such as expectations or probabilities) are desired, and the “biasing observable” for controlling the sampling process. A natural choice, which we use in the paper, is for the biasing and target observables to be the same. This is appropriate when the target observable takes continuous values.
>
> However, as discussed in the supplemental material (C4), a distinct biasing observable may lead to better algorithm convergence, while still aiding estimation if correlated with the target observable. For example, in AI safety, if we wish to estimate the probability of the binary target observable “the output will be safe (y/n)”, a distinct biasing observable that is continuous but correlated to the target will lead to superior convergence of the method.
>
> Good choices of these observables will require domain-specific knowledge. In the revision we will clarify how our approach can be tailored to users aims.
>
> ***(ii) Comment: “[…] extending analysis to larger models with pretraining sets [...] should also be done to be more convincing.”***
>
> **Response**: We agree it is important to demonstrate the approach on larger models. In particular, it is important to show that when models are scaled up that (a) this method remains efficient, and (b) total computational costs are not prohibitive.
>
> Regarding (a) our aim is to show that the specific method presented here is more efficient for estimating rare event probabilities than direct sampling. While it is correct that we have shown this concretely for a small model, the large corpus of theoretical and computational work on importance sampling which demonstrates this efficiency in many other complex systems gives confidence that it will also hold for larger LLMs.
>
> Regarding (b), in Sec. 4 we argue that the method presented would be feasible for a large company to apply to a foundation model, in contrast to direct sampling.  For many (e.g. academic) researchers, such large computational resources are unavailable. There are many algorithmic improvements we leave for future work (as those discussed in Sect.7 and the appendices) that will further improve efficiency and allow the study of rare events for larger models with more modest compute budgets.
>
> ***(iii) Comment: “[…] the cases which the authors describe are weak cases in which the proposed framework would be useful [...] repetition penalty is a widely used method that is even accessible in libraries [...]”***
>
> **Response**: We agree that removing repetitive generations is available in standard libraries. However, it is less obvious how to filter out high-ARI trajectories, and such a functionality would not be standard in libraries. Our aim here was to show how samples of rare generations could be explored to understand their properties. This exploration can characterise rare generations in terms of more easily calculable and filterable observables that co-occur with the rare event. In the example in the paper, the method allows us to identify that extreme ARI cases coincidentally also had high repetition, and as such, for this observable, standard filtering for repetition is an easy way to prevent these undesirable high-ARI generations.
>
> The properties of text for rare events of other observables will be different however, and only a general framework as ours is guaranteed to work systematically. Once identified, one could imagine applying the same approach to more expensive, whole-generation observables (e.g. safety evaluations involving model calls) to find cheaper proxies to monitor to prevent undesirable outcomes.
>
> We will clarify that this is the methodology and thought process we are demonstrating.
>
> ***(iv) Comment: “How would you interpret ARI scores outside the human-corresponding age?”***
>
> **Response**: The ARI, a standard quantifier of readability, is a regression model trained on text manually labelled by reading age. As with any regression model, it can extrapolate to scores outside the range of the training data. While we would expect that an ARI of –4 corresponds to simpler text than an ARI of –3, the linear equivalence of ARI score and reading age is not guaranteed to hold in these regions.

---

> > ### Author Rebuttal · Reviewer_pVSq · 2026-04-05
> >
> > Thank you for the response. I have raised my score. Be sure to add the clarifications you mentioned.

---

> > > ### Author Response · Authors · 2026-04-08
> > >
> > > We again would like to thank you for your response and suggestions. We believe these suggestions will help to further improve the clarity of our paper.

---

### Decision · Program_Chairs · 2026-04-30

**Decision:**

Accept (spotlight)

**Comment:**

This paper presents an end-to-end framework for rare event analysis in LLMs, adapting statistical mechanics techniques (importance sampling, histogram reweighting, transition path sampling) to efficiently estimate probabilities of rare LLM behaviors. The framework is demonstrated on TinyStories.

The paper received strong positive marks from two reviewers. Reviewer LyuJ described it as "well written and easy to follow" and praised the "novel" conceptual framing "borrowing from statistical physics and computational chemistry." Reviewer pVSq gave an originality score of 4 (excellent) and called it a "well motivate work" that "attempts to offer a cohesive idea."

Both positive reviewers raised their scores after the rebuttal, with Reviewer LyuJ stating "I already liked the paper in its original form" and adjusting upward because the "author responses further assured me that their paper represents good, solid work."

The one dissenting reviewer chose not to engage in the rebuttal process. Given that this reviewer's concerns were similar to the concerns addressed in rebuttal from positive reviewers, I am recommending acceptance.